# Orient Anything V2:
# Unifying Orientation and Rotation Understanding

**Zehan Wang**[1,2*], **Ziang Zhang**[1*], **Jiayang Xu**[1], **Jialei Wang**[1],

**Tianyu Pang**[3†], **Chao Du**[3], **Hengshuang Zhao**[4], **Zhou Zhao**[1,2‡]

[1]Zhejiang University; [2]Shanghai AI Lab; [3]Sea AI Lab; [4]The University of Hong Kong

https://orient-anythingv2.github.io/

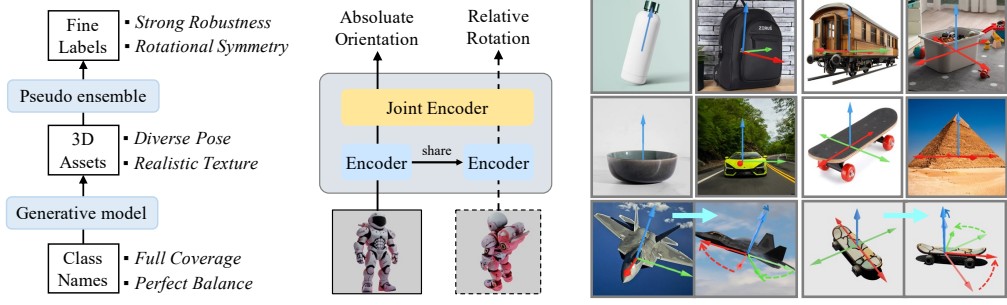

Figure 1: Overview of Orient Anything V2. We upgrade the foundation orientation estimation model from both **Data** and **Model** perspectives. It unifies the understanding of object orientation and rotation, achieving better estimation accuracy and gaining the **New Features** to handle rotational symmetry and relative rotation. Zoom in for the best view.

## Abstract

This work presents **Orient Anything V2**, an enhanced foundation model for unified understanding of object 3D orientation and rotation from single or paired images. Building upon Orient Anything V1, which defines orientation via a single unique front face, V2 extends this capability to handle objects with diverse rotational symmetries and directly estimate relative rotations. These improvements are enabled by four key innovations: **1)** Scalable 3D assets synthesized by generative models, ensuring broad category coverage and balanced data distribution; **2)** An efficient, model-in-the-loop annotation system that robustly identifies $0$ to $N$ valid front faces for each object; **3)** A symmetry-aware, periodic distribution fitting objective that captures all plausible front-facing orientations, effectively modeling object rotational symmetry; **4)** A multi-frame architecture that directly predicts relative object rotations. Extensive experiments show that Orient Anything V2 achieves state-of-the-art zero-shot performance on *orientation estimation*, *6DoF pose estimation*, and *object symmetry recognition* across 11 widely used benchmarks. The model demonstrates strong generalization, significantly broadening the applicability of orientation estimation in diverse downstream tasks.

---

[*]Equal Contribution.

[†]Project Leader.

[‡]Corresponding Author.

39th Conference on Neural Information Processing Systems (NeurIPS 2025).

# 1 Introduction

Estimating object orientation from images is a fundamental task of computer vision. 3D object orientation information plays crucial roles in robot manipulation [49, 37, 22], autonomous driving [25], AR/VR [11, 30, 4, 57], and spatial-aware image understanding [21, 28, 60] and generation [56, 50, 35].

Orient Anything V1 [48] is a foundation model for estimating the object orientation aligned with an object's unique front face. While it exhibits strong robustness and accuracy in absolute orientation estimation, it lacks an understanding of rotation (despite its intrinsic link to orientation). This deficiency results in difficulties handling numerous rotationally symmetric objects (simply classifying them as having no front face) and understanding object rotation relative to a specified reference frame. These limitations around rotation understanding restrict its utility in many downstream tasks.

In this work, we aim to develop an enhanced orientation estimation model, Orient Anything V2, with stronger generalization and deeper understanding of both object orientation and rotation. Our contributions include a scalable data engine and a more elegant model framework.

From the *data* perspective, Orient Anything V1 uses advanced VLM [12, 31] to annotate real 3D assets from Objaverse [8, 7]. Building on this data-driven motivation, we leverage advanced 3D generation models [51, 61] to further speed up data scaling-up and improve the data coverage and balance. Additionally, we assemble pseudo labels predicted by the V1 model across multi-view renderings and refine them through model-in-the-loop calibration. The proposed data engine enables highly cost-effective and flexible data scaling up, delivers robust annotation performance, and shows a strong understanding of rotationally symmetric objects. Our final dataset includes 600K assets, $12\times$ larger than the existing orientation dataset, with significantly higher annotation quality, accurately identifying 0 to N valid front faces.

From the *model* perspective, we first propose symmetry-aware orientation distribution, explicitly teaching the model to capture and predict rotational symmetry. Moreover, our model supports multi-frame input to directly predict relative rotations between frames. This design effectively bridges the knowledge transfer between absolute orientation and relative rotation, showing strong potential in reference-known scenarios.

Our experiments demonstrate the enhanced and novel capabilities of our model. It achieves superior performance on zero-shot orientation estimation and sets new records on zero-shot rotation estimation (i.e., 6DoF pose estimation [49, 26]), while also accurately handling and predicting different rotational symmetries.

To summarize, we propose Orient Anything V2, which improves Orient Anything V1 as follows:

- We propose a data engine that cost-efficiently scales up 3D asset collection and robustly annotates the 0 to N valid front faces to capture different object rotational symmetries.
- We introduce symmetry-aware distribution fitting as a learning objective, allowing the model to directly predict all plausible object orientations.
- We extend the model architecture to support multi-frame input, enabling it to directly estimate relative object rotations over the reference frame.
- Our model demonstrates strong zero-shot generalization across absolute orientation estimation, relative rotation estimation, and object symmetry recognition.

# 2 Related Work

## 2.1 Object Rotational Symmetry

Rotational symmetry [36, 34] indicates that an object may retain its original shape after being rotated by certain angles. This property is commonly found across various objects. Understanding an object's rotational symmetry is critical for 3D object recognition and generation [24, 59], pose estimation [17, 6], and robotic manipulation [39]. While some existing works [40] attempt to detect 3D rotational symmetry from single-view 2D images, they are constrained by limited training data and lack zero-shot generalization to open-world scenarios.

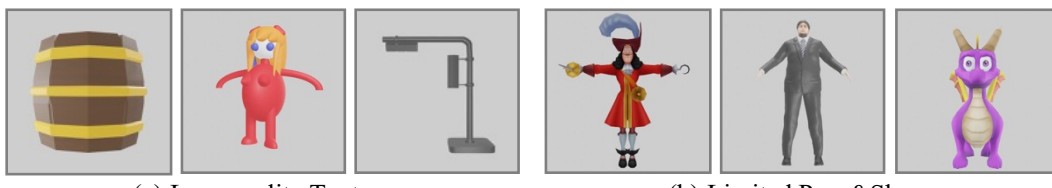

(a) Low-quality Texture            (b) Limited Pose&Shape

Figure 2: Real assets from Objavese suffer from (a) low-quality texture and (b) limited realism.

Our focus is on object orientation relative to a semantic "front" face. The number of possible valid front-facing orientations an object possesses is determined by its rotational symmetry around its vertical axis. For example, 180-degree symmetry means there are two distinct valid front faces. Objects with continuous rotational symmetry (symmetric at any angle), like balls, are considered to have no meaningful direction. In this work, we broaden the applicability of orientation estimation models by enabling the prediction of an object's azimuthal symmetry from a single 2D image. Our model demonstrates impressive zero-shot rotational symmetry recognition performance.

## 2.2 Relative Rotation Estimation

Predicting an object's rotation in the query frame relative to the reference frame is a fundamental capability in 6DoF pose estimation [26, 14, 55] and is crucial for robotics applications. Early methods [23, 18, 9] focused on specific instances or object categories. More recent approaches like OnePose [44] and OnePose++ [15] estimate object rotation by solving 2D-3D correspondences across views. POPE [10] follows a similar idea and achieves zero-shot rotation estimation with a single reference frame with the help of SAM [19] and DINOv2 [33]. However, the reliance on pixel matching makes these methods prone to failure under large viewpoint changes.

In contrast, we propose a purely implicit learning approach. Leveraging the inherent coupling between rotation and orientation, we extend the orientation estimation model to support multi-frame inputs, enabling direct zero-shot relative rotation prediction between arbitrary views.

## 2.3 Single-view Orientation Estimation

Estimating an object's 3D front-facing orientation (interpreted as its rotation relative to the canonical front view) from a single view, requires the model to have an inherent understanding of different objects' standard poses and front-facing appearances. Earlier works [53, 46, 42] are mainly limited to a small number of categories or specific domains. More recently, ImageNet3D [29] introduce a large-scale dataset with manually annotated 3D orientations. Orient Anything [48] achieves robust orientation estimation for any objects in any scenes by leveraging an advanced automated annotation pipeline, improved learning objectives, and real-world knowledge from the pre-trained vision model.

In this work, we further address several limitations of Orient Anything and upgrade the orientation estimation model from both data-driven (novel and scalable data engine) and model-driven (direct symmetry and rotation prediction) perspectives, resulting in Orient Anything V2.

# 3 Revisiting Orient Anything V1

Orient Anything V1 pioneers zero-shot object orientation estimation from single images. It introduces a VLM-based pipeline to annotate front faces of Objaverse 3D assets [8, 7], learns orientation estimation from their renderings via distribution fitting. It also provides a confidence score to indicate whether an object has a unique front face. To further advance the orientation prediction foundation model, we first dig into the potential limitations in its training data and framework.

**Disadvantages of Real 3D Assets**    1) *Imbalanced Category Distribution*: Stemming from the human biases in asset creation, real 3D datasets, such as Objaverse [8, 7], suffer from significant class imbalance. Common categories like buildings and characters make up a large proportion, while

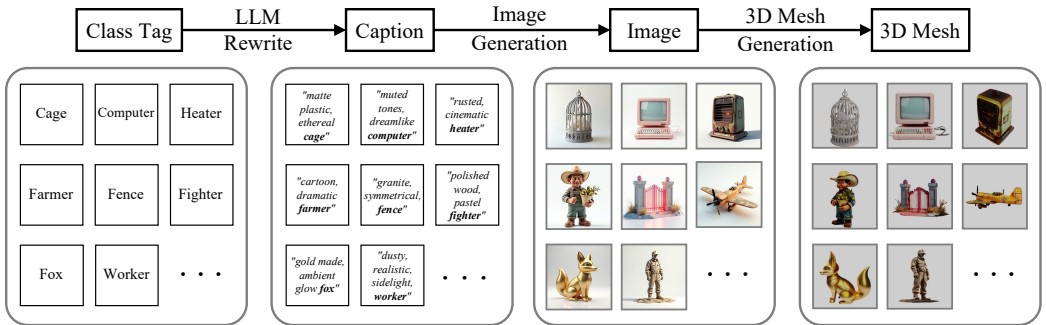

Figure 3: Overview of 3D Asset Synthesis Pipeline. We begin with class tags and use a series of advanced generative models to progressively generate high-quality 3D assets.

others, like uncommon animals, are severely underrepresented. 2) *Inconsistent Data Quality*: Current large-scale 3D datasets often lack high-quality assets with complete geometry and rich surface details. (Fig. 2 a) Moreover, many human-created meshes exhibit fixed poses, leading to a substantial domain gap from real-world object variations (Fig. 2 b).

**Limitations of Object Rotation Understanding**    1) *Ignored Rotational Symmetries*: Orient Anything V1 defines orientation based on the single, unique front face, overlooking the different rotational symmetries (i.e., multiple valid "front" faces). For the many symmetric objects in real world, the model cannot effectively distinguish or identify their potential orientations. 2) *Unsupported Relative Rotations*: The relative rotation between two views and the front-facing orientation (essentially the rotation relative to the front view) are inherently coupled. However, estimating relative rotation through independent absolute orientation predictions suffers from significant error accumulation, causing Orient Anything V1 to often fail in relative rotation estimation.

# 4    Scalable Data Engine

## 4.1    3D Asset Synthesis

Motivated by the recent remarkable progress in generative models and the successful application of synthetic data in downstream tasks [45, 56], we explore whether *synthetic 3D assets* can serve as scalable, high-quality data sources for orientation learning. To fully harness modern generative models, we construct our asset synthesis pipeline as a structured process: *Class Tag → Caption → Image → 3D Mesh*, as detailed below:

*Step 1: Class Tag → Caption.* To ensure broad category coverage and diversity, we follow SynCLR's approach [45], starting from ImageNet-21K [38] category tags, and use Qwen-2.5 [54] to generate rich captions that describe detailed object attributes and diverse poses. *Step 2: Caption → Image.* We use the state-of-the-art text-to-image model, FLUX.1-Dev [20], to generate images following the captions. Besides, we enhance captions with positional descriptors to promote explicit 3D structure and upright pose. *Step 3: Image → 3D mesh.* We employ the leading open-source image-to-3D model, Hunyuan-3D-2.0 [61], to produce high-quality 3D meshes from the synthesized images.

Finally, we generate 600k 3D assets in total, with approximately 30 items for each class tag in ImageNet-21K. These assets feature complete geometry, detailed textures, and balanced category coverage. In terms of scale, the new synthetic dataset is 12× larger than the filtered real dataset used in Orient Anything V1.

## 4.2    Robust Annotation

Orient Anything V1 employs VLM to annotate the unique canonical front view of 3D assets. However, this approach is limited by the VLM's underdeveloped spatial perception ability and struggles to handle diverse rotational symmetries. To address these challenges, we introduce a more effective and robust system for annotating 3D asset orientations.

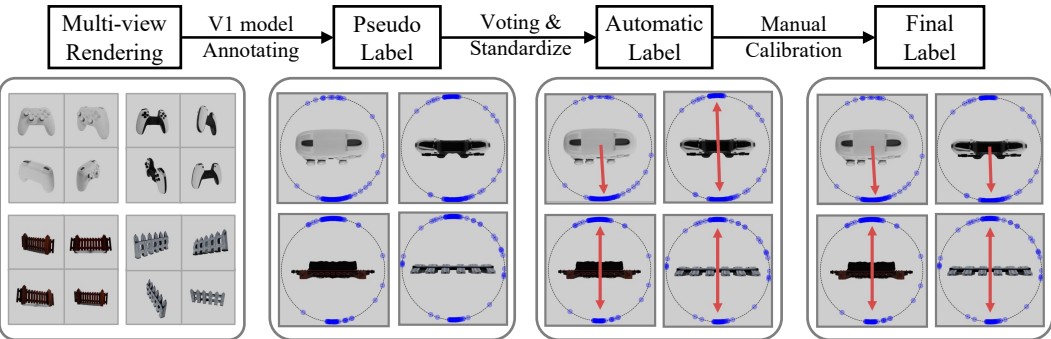

Figure 4: Overview of Robust Annotation Pipeline. "Pseudo Label" visualizes the azimuth direction of pseudo labels and objects in the horizontal plane. By fitting the pseudo labels to standard periodic distribution, we can robustly derive the orientation and symmetry label. Human calibration is only required for categories with symmetry inconsistencies.

**Intra-asset Ensemble Annotation**    We first train an improved orientation estimation model as our automatic annotator, based on the Orient Anything V1 paradigm and incorporating the additional real-world orientation dataset, ImageNet3D. Next, for each 3D asset, we employ this model to produce pseudo-labels for various renderings. Finally, we project these pseudo-labels, obtained from different viewpoints, back into a canonical 3D world coordinate system.

As shown in Fig. 4, the overall distribution of pseudo-labels on the horizon plane clearly indicates the object's possible orientations. To capture the main direction and rotational symmetries, we first arrange the discrete predicted azimuth angles over $[0°, 360°)$ into a probability distribution $\mathbf{P}_{\text{pseudo}} \in \mathbb{R}^{360}$. This distribution is then fitted to a periodic Gaussian distribution using the least squares method:

$$(\bar{\varphi}, \bar{\alpha}, \bar{\sigma}) = \arg\min_{\varphi,\alpha,\sigma} \sum_{i=0}^{359} \left( \mathbf{P}_{\text{pesudo}}(i) - \frac{\exp\left(\frac{\cos(\alpha(i-\varphi))}{\sigma^2}\right)}{2\pi I_0 \left(\frac{1}{\sigma^2}\right)} \right)^2 \tag{1}$$

where $\bar{\sigma}$ is the fitted variance. The phase $\bar{\varphi} \in [0°, 360°)$ represents the main azimuth direction. The periodicity $\bar{\alpha} \in \{1, 2, \ldots, N\}$ signifies $360/\bar{\alpha}$-degree rotational symmetry, possessing $\bar{\alpha}$ valid front faces, while $\bar{\alpha} = 0$ indicates no dominant orientation.

Ensembling multiple pseudo labels in the 3D world effectively suppresses outlier errors from single-view predictions, resulting in significantly more reliable annotations.

**Inter-assets Consistency Calibration**    Building on the rotational symmetry and orientation annotations for individual assets, we further perform human-in-the-loop consistency calibration across assets. Specifically, since our 3D assets are generated based on object category tags, they are naturally grouped by category. We assume that objects of the same category should share the same type of rotational symmetry. Based on this assumption, we analyze the annotated rotational symmetries within each category. If all assets within the same category demonstrate the same symmetries, we directly consider the annotations to be correct. If inconsistencies are found, we manually review all assets in that category to re-annotate or filter out incorrect annotations.

As each asset is annotated independently, the cross-asset consistency check and manual calibration offer an orthogonal perspective that efficiently and effectively enhances annotation reliability. Statistically, across 21k source category tags, we observe only minor inconsistencies in around 15% of categories, each involving a small number of assets. The finding further validates the accuracy and robustness of our ensemble annotation strategy.

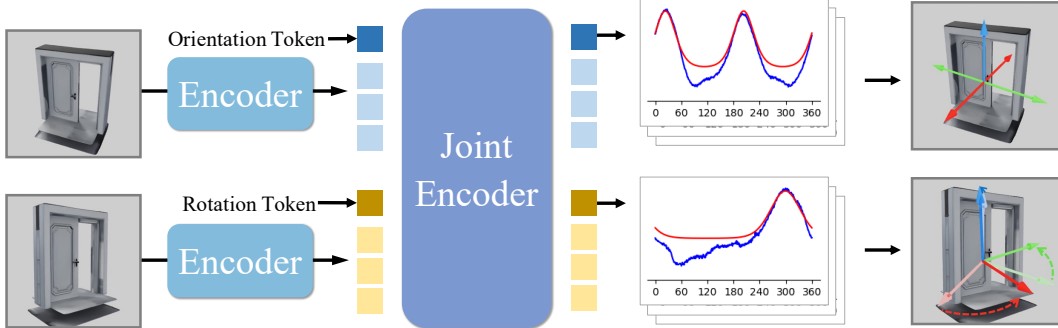

Figure 5: Framework of Orient Anything V2. One or two input frames are tokenized by DINOv2 and then jointly encoded using transformer blocks. We finally employ MLP heads to predict the orientation or rotation distributions from the encoded learnable tokens of each frame.

# 5 Framework

## 5.1 Symmetry-aware Distribution

Orient Anything V1 proposes an orientation distribution fitting task that guides the model to learn circular Gaussian distributions over azimuth, polar, and in-plane rotation angles, that preserve the similarity between neighboring angles. Each angle is modeled with a unimodal target distribution centered on a unique front-facing orientation. For symmetric objects with multiple or no semantic front faces, the model additionally predicts a low orientation confidence to filter them out.

To recognize different types of rotational symmetry and enable general orientation prediction for objects with multiple front faces, we further introduce the symmetry-aware periodic distribution as the training target. As discussed in Sec. 4.2, our ensemble annotation and consistency calibration approach enables accurate and robust labeling of 0 to N valid front-facing directions over the horizontal plane. To incorporate these annotations into prediction, we directly model 0 to N valid front faces within the azimuth angle distribution. This design naturally replaces V1's extra orientation confidence design. Instead, different kinds of rotational symmetries are captured directly from the predicted probability distribution. This more elegant framework enables the model to inherently share knowledge across all object categories.

For training, the target $\mathbf{P}_{azi} \in \mathbb{R}^{360}$ for the azimuth angle, originally represented as a circular Gaussian distribution, is adapted to be periodic:

$$\mathbf{P}_{azi}\left(i|\bar{\varphi}, \bar{\alpha}, \sigma\right) = \frac{\exp\left(\frac{\cos(\bar{\alpha}(i-\bar{\varphi}))}{\sigma^2}\right)}{2\pi I_0\left(\frac{1}{\sigma^2}\right)} \tag{2}$$

where $\bar{\varphi}$ and $\bar{\alpha}$ are the phase (azimuth angle) and periodicity (rotation symmetry) fitted from Sec. 4.2, $\sigma$ is the variance hyper-parameter, and $i = 0°, \ldots, 359°$ is the angle index. Target probability distributions for the polar angle $\mathbf{P}_{pol} \in \mathbb{R}^{180}$ and in-plane rotation angle $\mathbf{P}_{rot} \in \mathbb{R}^{360}$ are constructed using a similar method, but without the periodicity parameter.

During inference, the predicted angle distributions are fitted to a standard distribution model using the least squares method, similar to Eq. 1. The resulting parameters (azimuth periodicity $\hat{\alpha}$, azimuth angle $\hat{\varphi}$, polar angle $\hat{\sigma}$ and rotation angle $\hat{\delta}$), directly indicate the object's $\hat{\alpha}$ valid front faces (i.e., symmetric with $360/\alpha$ degree rotation) and their corresponding front-facing directions in 3D space.

## 5.2 Relative Rotation Estimation

To establish a connection between absolute orientation and relative rotation, enabling knowledge sharing and transferring, we modify the network architecture to support dynamic inputs from one or multiple images.

As shown in Fig. 5, we mainly follow VGGT [47], first using a visual encoder, DINOv2 [33], to encode each input image into $K$ tokens, augmented with learnable tokens. The combined set of

| Model | SUN-RGBD | | ARKitScenes | | Pascal3D+ | | Objectron | | ImageNet3D[†] | | Ori_COCO |
|---|---|---|---|---|---|---|---|---|---|---|---|
| | Med↓ | Acc30°↑ | Med↓ | Acc30°↑ | Med↓ | Acc30°↑ | Med↓ | Acc30°↑ | Med↓ | Acc30°↑ | Acc↑ |
| OriAny.V1 | 33.94 | 48.5 | 77.58 | 35.8 | 22.90 | 55.0 | 30.67 | 49.6 | **13.34** | **71.3** | 72.4 |
| OriAny.V2 | **26.00** | **55.4** | **36.48** | **43.2** | **15.02** | **72.7** | **22.62** | **56.4** | 15.26 | 65.2 | **86.4** |

Table 1: Zero-shot Absolute Orientation Estimation. [†]: ImageNet3D is used for training Orient Anything V2. To ensure a fair comparison, the compared V1 model is fine-tuned on ImageNet3D. Best results are highlighted in **bold**.

tokens from all frames is then passed into a unified transformer block. The final learnable token corresponding to each frame is used for prediction. Specifically, the learnable token for the first frame is initialized differently and is used to predict the absolute orientation using the symmetry-aware distribution described in Sec. 5.1. Tokens from subsequent frames predict the object rotation relative to the first frame through a similar probability fitting task, but without considering symmetry.

### 5.3 Training Setting

Our model is initialized from VGGT, a large feed-forward transformer with 1.2 billion parameters pre-trained on 3D geometry tasks. We repurpose its original "camera" token, designed to predict camera extrinsics, to predict object orientation and rotation. This leverages the inherent correlation between camera pose and object rotation. We train the model to fit target orientation (or rotation) distributions using Binary Cross-Entropy (BCE) loss for 20k iterations. A cosine learning rate scheduler is used with an initial rate of 1e-3. Input frames are resized to 518, and random patch masking is used for data augmentation to simulate real-world occlusion. The effective batch size is set to 48, where 1-2 frames are randomly sampled for each training sample. The training dataset comprises the ImageNet3D training set and newly collected 600k synthetic assets. Furthermore, we observe that most objects exhibit only four types of rotational symmetry: $\{0, 1, 2, 4\}$. Therefore, we restrict our training to consider only these four cases. Any fitted periodicity $\bar{\alpha} > 4$ is mapped to 0.

## 6 Experiment

### 6.1 Zero-shot Orientation Estimation

**Benchmark & Baselines** Predicting the 3D orientation of objects from a single image is our core focus. We mainly compare with Orient Anything V1 [48] on ImageNet3D [29] test set and unseen test datasets, SUN-RGBD [41], ARKitScenes [3], Pascal3D+ [52], Objectron [1] and the Ori_COCO [48]. Since current testing datasets often provide only one ground truth orientation, even for symmetric objects, when Orient Anything V2 predicts multiple orientations, we simply select the one closest to facing the camera as the prediction. The main evaluation metrics are the median 3D angle error (Med↓) and accuracy within 30 degrees (Acc30°↑). For Ori_COCO, where 20 samples are collected for each class and annotated within 8 horizontal orientations, recognition accuracy (Acc↑) is used.

**Main Results** In Tab. 1, we present the comparative results on single view-based orientation estimation. Overall, Orient Anything V2 significantly improves upon V1, benefiting from diverse synthetic data and robust ensemble annotation. On the representative Ori_COCO benchmark, our method achieves 86.4% accuracy and performs well on categories where V1 struggled, such as bicycles. Achieving state-of-the-art results on numerous real-world image datasets highlights our method's generalization ability.

### 6.2 Zero-shot Rotation Estimation

**Benchmark & Baselines** We benchmark zero-shot 6DoF object pose estimation performance under a single reference view. Evaluation is conducted on four widely used datasets: LINEMOD [16], YCB-Video [5], OnePose++ [15], and OnePose [44]. Objects are prepared using the cropping and matching, following [10]. Comparisons are made against three state-of-the-art zero-shot 6DoF object pose estimation methods: Gen6D [27], LoFTR [43], and POPE [10]. Standard metrics for relative object pose estimation are used: median error (Med) and accuracy within 15° and 30° (Acc15 and Acc30), computed for each sample pair.

| Model | LINEMOD | | | YCB-Video | | | OnePose++ | | | OnePose | | |
|---|---|---|---|---|---|---|---|---|---|---|---|---|
| | Med↓ | Acc30↑ | Acc15↑ | Med↓ | Acc30↑ | Acc15↑ | Med↓ | Acc30↑ | Acc15 | Med↓ | Acc30↑ | Acc15↑ |
| *POPE's Sampling (Average rotation angle: 14.85°)* | | | | | | | | | | | | |
| Gen6D | 44.86 | 36.4 | 9.6 | 54.48 | 23.2 | 7.7 | 35.43 | 41.1 | 15.8 | 17.78 | 89.3 | 38.9 |
| LoFTR | 33.04 | 56.2 | 32.4 | 19.54 | 68.6 | 47.8 | 9.01 | 89.1 | 70.3 | 4.35 | 96.3 | 91.8 |
| POPE | 15.73 | 77.0 | 48.3 | 13.94 | 80.1 | 54.4 | 6.27 | 89.6 | 72.8 | **2.16** | 96.2 | 91.1 |
| OriAny.V2 | **7.82** | **98.07** | **89.7** | **6.07** | **91.6** | **86.4** | **6.18** | **99.7** | **96.6** | 6.76 | **99.7** | **95.7** |
| *Random Sample (Average rotation angle: 78.22°)* | | | | | | | | | | | | |
| POPE | 98.03 | 10.3 | 4.3 | 41.88 | 40.9 | 27.2 | 88.21 | 25.6 | 19.8 | 45.73 | 45.1 | 37.3 |
| OriAny.V2 | **28.83** | **51.6** | **28.3** | **15.78** | **61.2** | **48.7** | **12.83** | **85.5** | **58.8** | **11.72** | **86.7** | **63.4** |

Table 2: Zero-shot Relative Rotation Estimation (i.e., pose estimation with one reference view). We evaluate two strategies for sampling query-reference view pairs: (1) query-reference pairs provided by POPE [10], and (2) randomly sampled pairs. The average rotation angles between views for each sampling strategy are 14.85° and 78.22°, respectively.

**Main Results** Tab. 2 includes zero-shot two-view relative rotation estimation results compared with state-of-the-art pose estimation methods. With small relative rotations (using POPE's sampling), our model achieves the overall best performance across the four datasets. More importantly, our method's advantage is significantly larger when the relative rotation between the query and reference frame is larger (using random sampling). The significant performance drop of the previous method stems from the reliance on explicit feature matching, which becomes unreliable with large rotations due to less view overlap and scarcer reliable matching points. In contrast, our approach understands images from different viewpoints by considering overall meaning rather than just detailed matching. This makes it more robust to challenging large rotations.

## 6.3   Zero-shot Symmetry Recognition

**Benchmark & Baselines** We assess our method's zero-shot performance in predicting object rotational symmetry in the horizontal plane. This evaluation uses the recent, large-scale 3D object datasets with rotational symmetry annotations: Omni6DPose [58], which contain 149 distinct object classes. To ensure their orientation definition aligns with our front-facing direction, we manually select a subset of 3-5 assets per category and render 2 views per 3D asset for testing. This resulted in 838 testing sample. During inference, models receive a single rendering and predict the four kinds of rotational symmetry predictions. As there are currently no dedicated zero-shot models for predicting object rotational symmetry from a single view, we employ advanced VLMs (Qwen2.5VL-72B [2], GPT-4o [31], GPT-o3 [32], and Gemini-2.5-pro [13]) as baselines. We evaluate their ability to predict horizontal plane rotational symmetry using a multiple-choice format, with recognition accuracy as the metric.

**Main Results** We present a comparison of our method against various advanced general VLMs for identifying object horizontal rotational symmetry in Tab. 3. Our results indicate that recognizing object rotational symmetry is a challenging problem even for the strongest VLMs, thereby limiting their ability to fully understand the 3D spatial state from 2D images. In contrast, benefiting from high-quality annotations and a unified learning objective, our model achieves 65% accuracy in distinguishing object rotational symmetry. Combining this strong symmetry recognition ability alongside the robust and accurate absolute orientation estimation performance demonstrated in Sec. 6.1, our model can accurately infer multiple potential orientations from a single image in real applications.

| | Omni6DPose |
|---|---|
| | Acc↑ |
| Random | 25.0 |
| Qwen2.5VL-72B | 55.8 |
| Gemini-2.5-pro | 44.4 |
| GPT-4o | 62.5 |
| GPT-o3 | 53.7 |
| OriAny. V2 | **65.2** |

Table 3: Zero-shot horizontal rotational symmetry recognition.

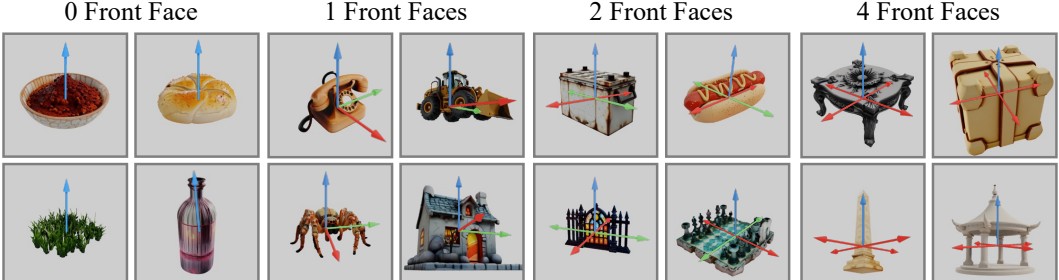

Figure 6: Visualization of synthetic 3D assets and robust annotation.

| Row | Assets Type | Assets Number | Initialized Weights | Orientation Estimation | | | Rotation Estimation | | | |
|---|---|---|---|---|---|---|---|---|---|---|
| | | | | Objectron | | Ori_COCO | LINEMOD | | YCB-Video | |
| | | | | Med↓ | Acc15↑ | Acc↑ | Med↓ | Acc15↑ | Med↓ | Acc15↑ |
| 1 | Real | 40K | VGGT | 25.05 | 54.3 | 74.8 | 10.70 | 69.8 | 15.49 | 72.5 |
| 2 | Synthetic | 40K | VGGT | 24.44 | 54.6 | 74.6 | 10.16 | 74.1 | 7.28 | 76.2 |
| 3 | Synthetic | 200K | VGGT | 23.82 | 55.0 | 75.2 | 10.22 | 74.5 | 7.49 | 78.6 |
| 4 | Synthetic | 400K | VGGT | 25.09 | 53.9 | 75.4 | 9.78 | 76.3 | 6.48 | 80.5 |
| 5 | Synthetic | 600K | VGGT | 22.62 | 54.8 | 86.4 | 7.82 | 89.7 | 6.07 | 84.7 |
| 6 | Synthetic | 600K | DINOv2 | 26.70 | 52.6 | 79.0 | 15.11 | 49.1 | 13.78 | 52.6 |
| 7 | Synthetic | 600K | None | 62.08 | 13.5 | 25.6 | 16.54 | 45.3 | 13.93 | 52.2 |

Table 4: Ablation study. For the rotation estimation, we employ POPE's sampling pairs.

## 6.4 Ablation Study

**Quality of Synthetic 3D Assets** Fig. 6 visualizes our synthetic dataset and the labelled orientation, qualitatively demonstrating the high quality of both the synthetic data and its annotations. Quantitatively, Rows 1 and 2 of Tab. 4 show the comparison of training with an equal amount of annotated real or synthetic 3D assets. We observe that both data sources yield comparable results for absolute orientation estimation. However, for rotation estimation (on LINEMOD and YCB-Video), training with synthetic assets provides a significant advantage. This may be because synthetic assets possess richer, more realistic textures, which are more crucial for understanding rotation.

**Effect of Scaling Data** In Rows 2, 3, 4, and 5 of Tab. 4, we explore the impact of data scale on the performance of final orientation and rotation estimation. Overall, with the same training step, encountering more diverse data and 3D assets during training leads to better overall performance. Specifically, we find that rotation estimation is more sensitive to data scale than orientation estimation. This may be because orientation relies on overall semantics and structure, while rotation estimation requires understanding diverse textures and fine-grain details to capture cross-view relationships.

**Effect of Geometry Pre-training** Tab. 4 (Rows 5-7) presents our experiments of different model initialization strategies. Training without any pre-trained initialization yields the worst results. Initializing the separated visual encoder with DINOv2 introduces valuable high-quality semantic and object structure information, leading to substantial performance gains. We observe further improvements in rotation estimation by using VGGT, pre-trained specifically on 3D geometric tasks, which boosts the model's comprehension of object geometry.

## 7 Conclusion

We present Orient Anything V2, an advanced model for unified object orientation and rotation understanding. Through introducing the scalable data engine, a symmetry-aware distribution learning target, and a multi-frame framework, our model enables: 1) Stronger single-view absolute orientation estimation. 2) Advanced two-frame object relative pose rotation estimation. 3) Powerful object horizontal rotational symmetry recognition. In practice, the model can simultaneously and accurately predict multiple valid front faces of objects, making it well-suited for diverse objects and real-world application scenarios.

**Limitation**  While our models exhibit strong generalization to diverse in-the-wild objects in real images, we find that the inherent ambiguity of monocular images leads to less accurate predictions in views with very low information or severe occlusion. Furthermore, the current framework supports a maximum of two input frames. Extending the model to handle more frames will be an important direction for supporting video understanding applications.

## Acknowledgements

This work was supported in part by National Key R&D Program of China (No. 2022ZD0162000) and National Natural Science Foundation of China (No. 62222211, U24A20326, 624B2128, 62422606 and 62201484)

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

# A More Visualizations of Images in The Wild

In 7 8, 9, 10, 11, 12, 13, we present more visualizations of images from various domains containing different objects. In these images, our model shows strong abilities in single-view absolute orientation estimation, powerful object horizontal rotational symmetry recognition and two-frame object relative pose rotation estimation, further highlighting the impressive zero-shot capability of Orient Anything V2.

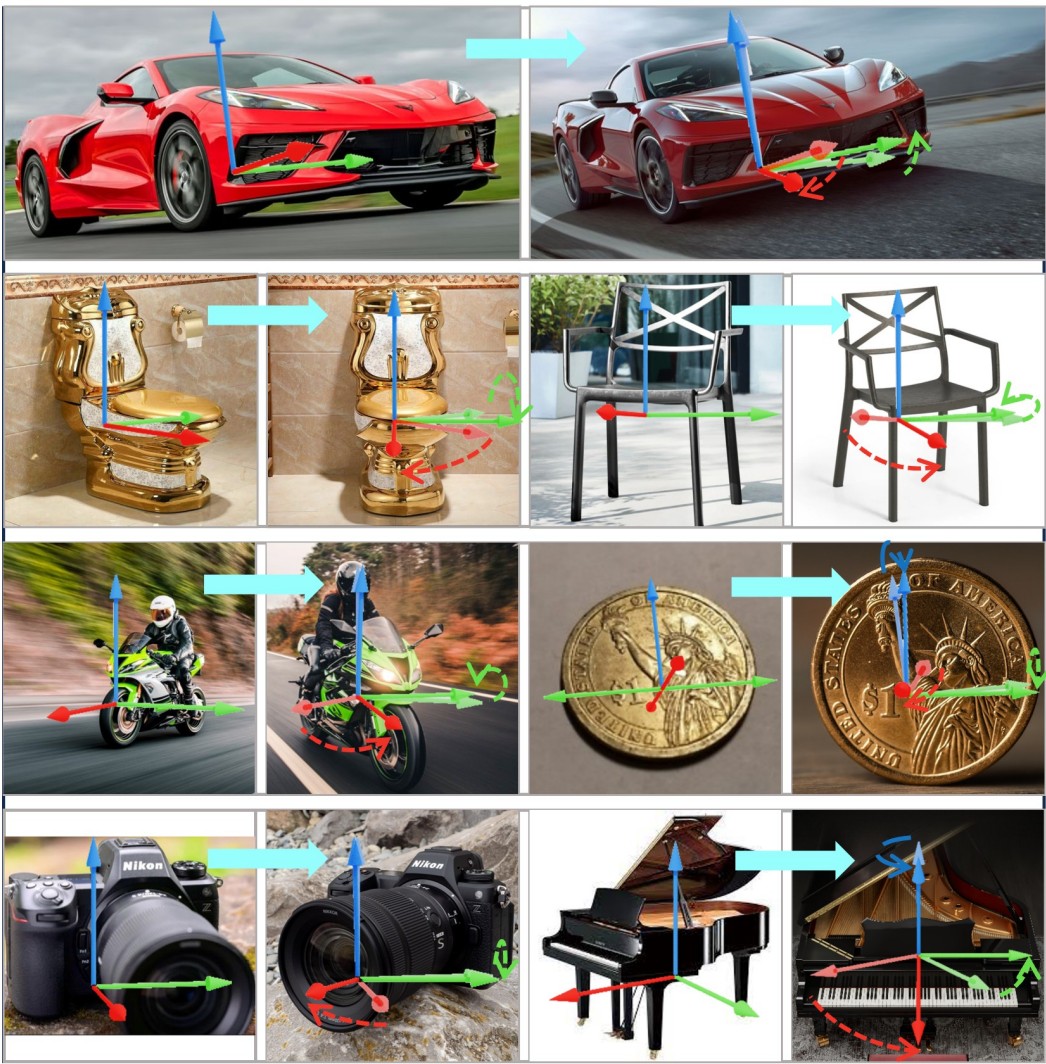

Figure 7: Relative pose rotation estimation for images in the wild.

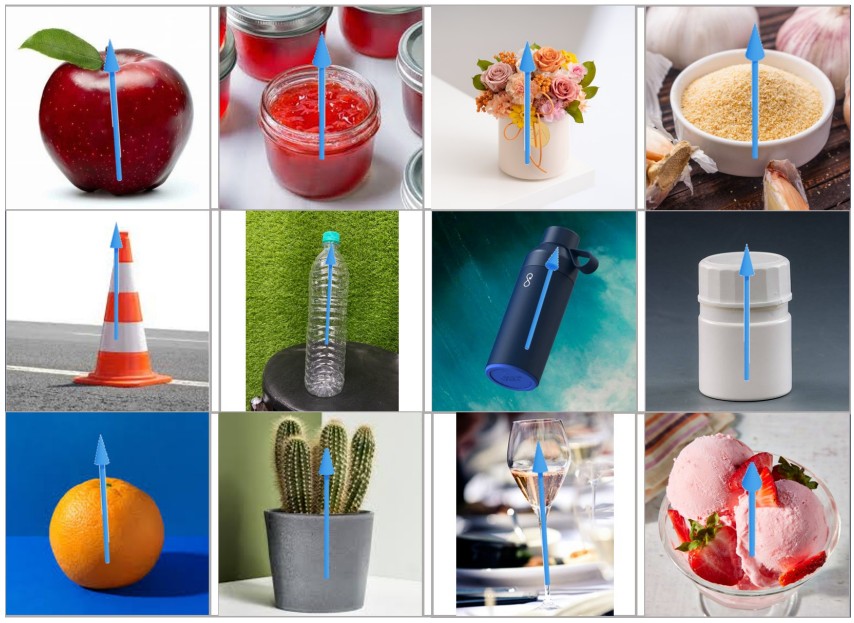

Figure 8: Orientation estimation and Rotational symmetry recognition results on objects has no front direction.

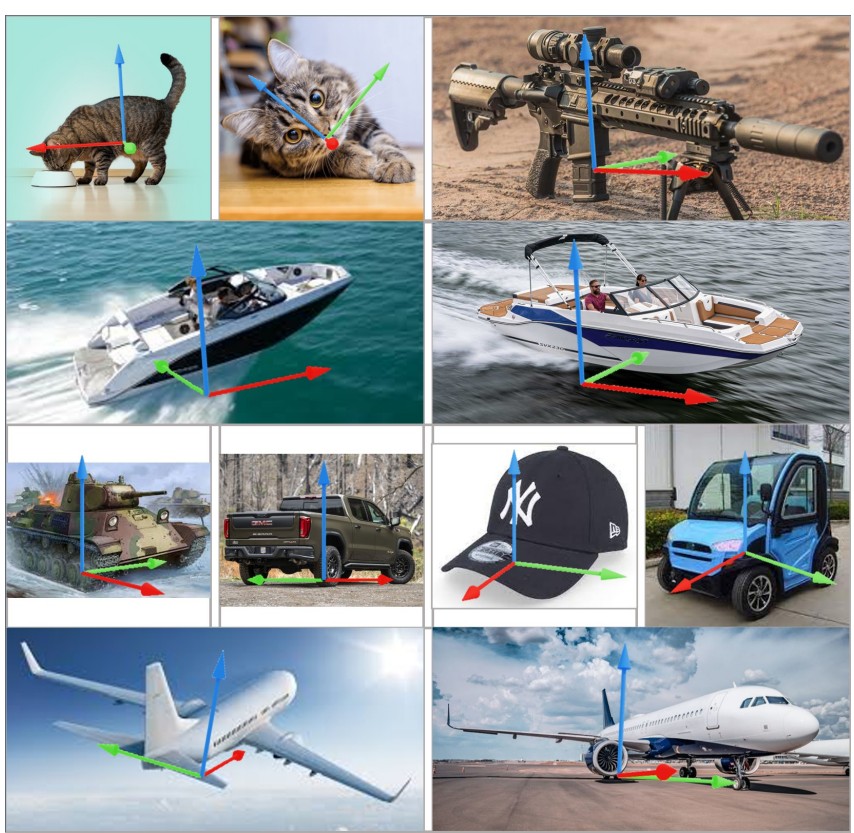

Figure 9: Orientation estimation and Rotational symmetry recognition results on objects has one front direction. Part 1.

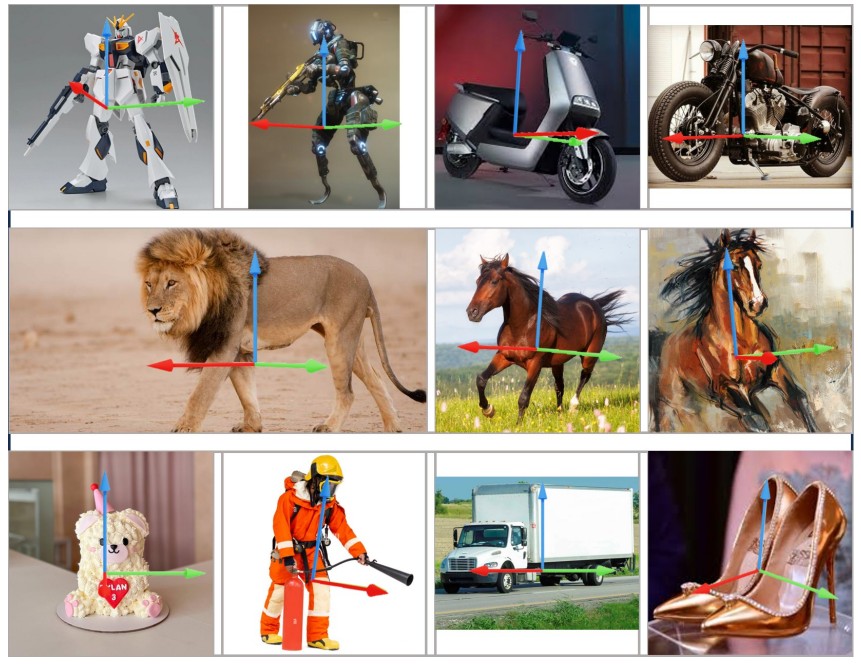

Figure 10: Orientation estimation and Rotational symmetry recognition results on objects has one front direction. Part 2.

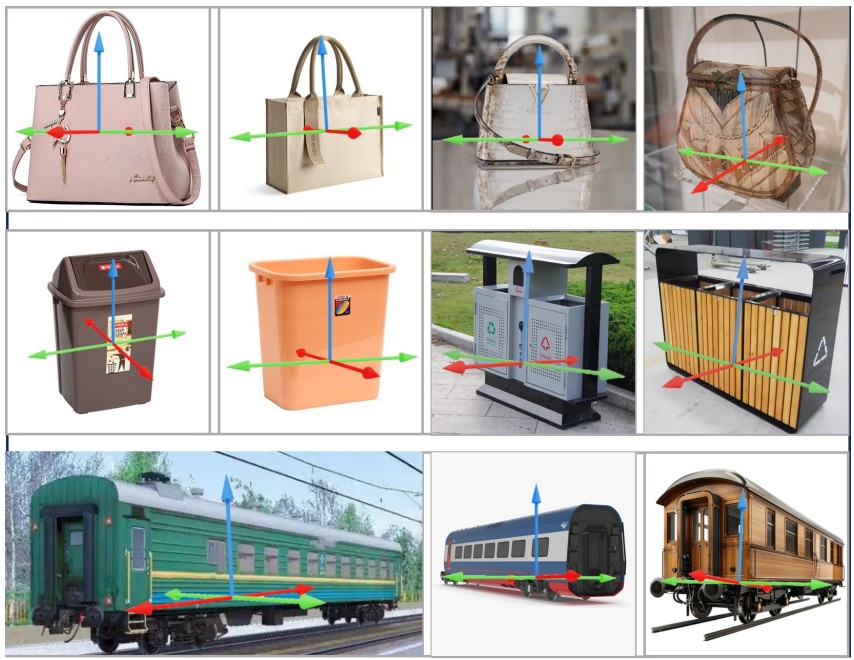

Figure 11: Orientation estimation and Rotational symmetry recognition results on objects has two front direction. Part 1.

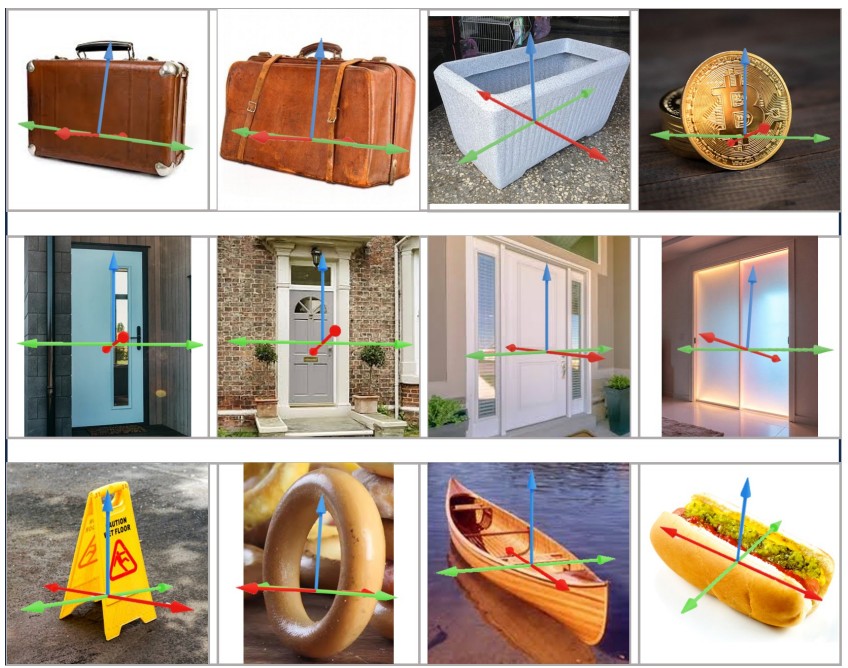

Figure 12: Orientation estimation and Rotational symmetry recognition results on objects has two front direction. Part 2.

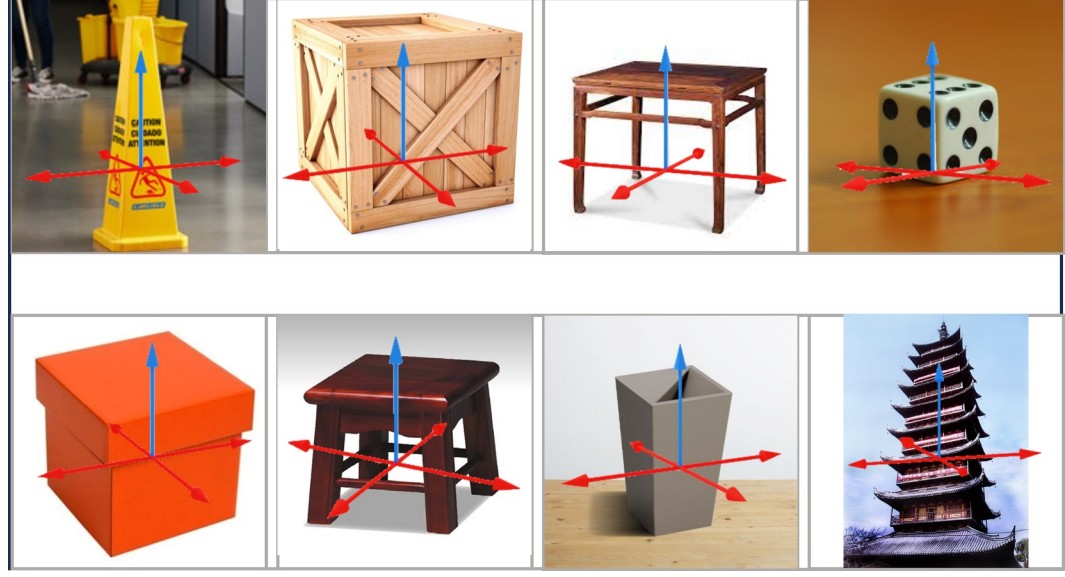

Figure 13: Orientation estimation and Rotational symmetry recognition results on objects has four front direction.

