# OpenReview forum: "Orient Anything V2: Unifying Orientation and Rotation Understanding"
_NeurIPS.cc/2025/Conference — NeurIPS 2025 spotlight_

### Official Review · Reviewer_tGZ7 · 2025-06-30

**Clarity:** 2
**Significance:** 3
**Originality:** 2
**Rating:** 5
**Confidence:** 4

**Summary:**

This paper presents Orient Anything V2, an upgraded foundation model for understanding 3D object orientation and rotation from single or paired images. Compared to its previous version, V2 claims: a new pipeline for generating high-quality synthetic 3D assets, a robust method to identify 0 to N valid front-facing directions per object, a symmetry-aware learning objective that directly models rotational symmetry, and a multi-frame architecture that supports relative rotation estimation. The proposed model achieves strong generalization across multiple real-world benchmarks for orientation estimation, 3D pose estimation, and rotational symmetry recognition.

**Questions:**

What is the best performance of Orient Anything V2 without using the VGGT pre-trained weights? Is it still comparable to the baseline?

**Ethical Concerns:**

["NO or VERY MINOR ethics concerns only"]

**Final Justification:**

I’ve read through all the reviews and the rebuttal carefully. Overall, I found the authors well responded to the points raised, and I appreciate the addition of new experiments (e.g., initializing with DINOv2 and showing the effect of the proposed data engine, etc). Therefore, I raised my score to accept.

I recommend the authors to carefully consider the comments and suggestions provided by the reviewers such as missing references/experiments, and address them for the final camera-ready version

**Limitations:**

Similar to its predecessor, Orient Anything V2 does not estimate object translation, which is a critical component for many real-world applications, such as robotics and AR/VR.

**Paper Formatting Concerns:**

The paper does not have any formatting concerns.

**Quality:**

2

**Strengths And Weaknesses:**

Strengths:
- The problem addressed in this paper, orientation estimation of arbitrary objects, is well motivated.
- The paper is overall clear and easy to follow.

Weaknesses:
- The paper overlooks several important related works that address similar tasks (pose estimation from a single image and orientation distribution prediction) such as RelPose [1] and NOPE [2].
- In Table 1 and its accompanying analysis, the authors claim that “Orient Anything V2 significantly improves upon V1, benefiting from diverse synthetic data and robust ensemble annotation“. This suggests that the improvement primarily stems from the dataset, rather than from the method design changes. This is an important point, as the model architecture and training losses appear very similar between V1 and V2. If true, the contribution of this work would lie mostly on the data side.
- In Table 4, the performance drops significantly when the model is not initialized with VGGT pre-trained weights, even falling below the baseline. This raises concerns about how much of the performance gain can truly be attributed to the proposed method itself.

---

> ### Author Rebuttal · Authors · 2025-07-31
>
> Thank you for your valuable review and suggestions. Below we respond to the comments in **Weaknesses (W)** and **Questions (Q)**.
>
> ---
>
> ## W1: Overlooks several important related works
>
> Thank you for your reminder. We will add discussions of these works and revise some of our statements in the updated version. While these works achieve a certain degree of open-world capability, their generalization to real-world scenarios remains limited by training scale and model capacity. We provide additional comparisons of median error (Med↓) below:
> (*since NOPE did not release open-source checkpoints, we failed to reproduce its results during rebuttal*)
>
> | Model               | LINEMOD   | YCB-Video | OnePose++ | OnePose   |
> | ------------------- | --------- | --------- | --------- | --------- |
> | **POPE's Sampling** |           |           |           |           |
> | POPE                | 15.73     | 13.94     | **6.27**  | **2.16**  |
> | VGGT                | 16.64     | 10.97     | 17.76     | 14.08     |
> | Relpose-co3dv1      | 16.74     | 13.53     | 14.54     | 11.37     |
> | Relpose-co3dv2      | 14.27     | 9.76      | 12.99     | 9.52      |
> | OriAny. V1          | 22.67     | 16.27     | 13.32     | 22.30     |
> | OriAny. V2          | **9.45**  | **6.53**  | 7.62      | 8.53      |
> | **Random Sampling** |           |           |           |           |
> | POPE                | 98.03     | 41.88     | 88.21     | 45.73     |
> | VGGT                | 83.96     | 20.75     | 52.18     | 40.30     |
> | Relpose-co3dv1      | 100.05    | 43.17     | 81.35     | 76.67     |
> | Relpose-co3dv2      | 95.49     | 31.21     | 83.21     | 47.48     |
> | OriAny. V1          | 86.63     | 43.30     | 26.88     | 75.19     |
> | OriAny. V2          | **32.08** | **13.78** | **12.94** | **16.29** |
>
> ---
>
> ## W2: Source of improvement
>
> Our improvements are demonstrated in three aspects:
> 1. Absolute orientation estimation (Table 1 and Sec 6.1),
> 2. Relative rotation estimation (Table 2 and Sec 3.2),
> 3. Rotation symmetry recognition (Table 3 and Sec 3.3).
>
> The dataset mainly contributed to the absolute orientation estimation in Table 1. However, the improvements in relative rotation estimation and rotation symmetry recognition are due to the updated model architecture (supporting multi-frame input) and learning objectives (periodic angular distributions).
>
> Therefore, across all three dimensions, both our new data engine and framework design are the main contributions.
>
> ---
>
> ## W3 & Q1: Usage of VGGT pre-training
>
> Firstly, regarding the baselines: **Orient Anything v1 is initialized with DINOv2, while POPE uses pre-trained DINOv2 and SAM**. Under a fair comparison setting with DINOv2 initialization, our method shows comprehensive advantages, as shown below:
>
> | Method                              | Objectron | Objectron | Ori_COCO | LINEMOD   | LINEMOD   | YCB-Video | YCB-Video |
> | ----------------------------------- | --------- | --------- | -------- | --------- | --------- | --------- | --------- |
> |                                     | Med↓      | Acc15↑    | Acc↑     | Med↓      | Acc15↑    | Med↓      | Acc15↑    |
> | POPE                                | -         | -         | -        | 15.73     | 48.30     | 13.94     | **54.4**  |
> | OriAny. V1                          | 30.67     | 49.6      | 74.8     | 22.67     | 31.10     | 16.27     | 45.80     |
> | OriAny. V2(DINOv2 initialized only) | **26.70** | **52.6**  | **79.0** | **15.11** | **49.07** | **13.78** | 52.65     |
>
> Secondly, we believe that strong pre-training is the foundation for developing powerful models. For instance, VGGT itself is initialized with DINOv2, Depth Anything v2 heavily relies on DINOv2, and Segment Anything uses MAE and CLIP pre-training. We believe it is normal for performance to drop without pre-training, and it's valuable to demonstrate how far a model can go when built upon existing research.
>
> ---
>
> ## Limitation 1: Does not estimate object translation
>
> Object translation in images relates to 2D bounding box and depth, which correspond to object detection and depth estimation models, respectively. Given the strong generalization and modularity of these methods, our work focuses solely on object orientation estimation. In practical applications such as robotics and AR/VR, one can modularly combine object detection, depth estimation, and our orientation estimation model to obtain the object's absolute pose, including both 3D translation and rotation.
>
> ---

---

> ### Comment · Reviewer_tGZ7 · 2025-08-05
> **Official comments by Reviewer tGZ7**
>
> Thanks to the authors for addressing all the concerns raised by myself and the other reviewers. I’ve read through all the reviews and the rebuttal carefully. Overall, I found the authors well responded to the points raised, and I appreciate the addition of new experiments (e.g., initializing with DINOv2 and showing the effect of the proposed data engine, etc). Therefore, I’m inclined to raise my score.

---

> > ### Author Response · Authors · 2025-08-05
> > **Thank you for your support!**
> >
> > We sincerely appreciate your kind support. In our final revision, we will further enhance the paper by incorporating the valuable insights gained from the rebuttal discussions. Thank you once again for your guidance. Please let us know if you have any further questions or suggestions; we would be pleased to address them.

---

### Official Review · Reviewer_krXG · 2025-07-02

**Clarity:** 3
**Significance:** 3
**Originality:** 3
**Rating:** 5
**Confidence:** 4

**Summary:**

This paper presets Orient Anything V2 - a model for estimating the absolute 3D orientation (the pose of an object defined by its front face) and relative rotation (3D transform between pose in two images) of objects in images. The main distinctions to Orient Anything V1 are the enabling direct relative pose estimation and handling (rather than ignoring) objects with symmetry.

To achieve this, the paper makes two significant contributions. First, a data engine is developed to generate a large amount of high quality 3D assets (ImageNet21K class label -> object description from LLM -> image generated by generative model -> 3D mesh generated by Hunyan 3D). These assets are then pseudo-labeled using the V1 model and these predictions are aggregated to determine symmetry and the number of front faces with a final manual consistency check. Given this large set of labeled assets, a transformer-based model is trained to jointly predict an orientation distribution relative to a world coordinate frame, and relative rotation distribution. The proposed approach achievers state-of-the-art absolute orientation estimation, relative rotation estimation, and symmetry recognition.

**Questions:**

- What is the effect of using synthesized 3D data rather than Objeverse? Is the 3D engine worth the effort?
- With respect to the sim2real domain gap, could the authors describe scenarios where the proposed approach may have critical failures?
- Could the authors describe, in as much detail as possible, how the world coordinate frame is chosen and why (is it the center of mass of the object, 3D bounding box center?) and how this is handled at inference time (for an arbitrary 2D image, how can we know were the point is in 3D for the coordinate system that we learned to predict in?)? What is the process to draw the arrows on the in-the-wild images?

**Ethical Concerns:**

["NO or VERY MINOR ethics concerns only"]

**Final Justification:**

I initially had a positive opinion of this work and asked for some clarifications that the authors addressed very well (justifying the data engine empirically, and clarifying how the visualizations in the paper are actually generated). I don't have concerns for this paper and maintain my positive rating.

**Limitations:**

The authors adequately address the weaknesses.

**Paper Formatting Concerns:**

No major formatting concerns.

**Quality:**

3

**Strengths And Weaknesses:**

**Strenghts**

Quality
- Extensive ablation studies - the paper ablates data scaling, model initialization, which are the main design components. As expected, rotation estimation benefits more from higher data scale.
- Extensive evaluation and strong performance - to justify the claim of referring to their model as a foundation model, the authors demonstrate its zero-shot generalization on orientation estimation, relative pose estimation, and symmetry recognition. Successfully unifying these tasks and obtaining strong performance is a significant achievement. Compared with prior work, the model has strong performance even on views with a large angle gap between the frame pairs.
- Model design - The architecture is simple and effective. Following the design of VGG-T and predicting a distribution over possible poses while handling periodicity for symmetry is elegant.

Clarity
- The paper is well written with high quality figures and easy to follow. Some technical and implementation details are missing that should be included (see weaknesses)

Originality
- Data engine - it is exciting to see that a combination of language models and image and 3D generative models can be thoughtfully combined to synthesize a dataset that is actually useful. Looking toward the future, leveraging generative models for data synthesis will be essential, and this work will inform future works attempting to do this.

Significance
- The significance is clear -- this work unifies three pose understanding tasks in a single model with good zero-shot performance. There are clear downstream applications in robotics or 3D content generation.

**Weaknesses**

Quality
- Understanding the effect of the data engine would greatly strengthen the paper. How much would performance reduce if instead of the data engine, Objaverse was used off-the-shelf?
- Given the significant reliance on synthetic data, this warrants an explicit discussion on the visual domain gap between synthetic data and test data, as well as what can be effective mitigation strategies.

Clarity
- How is the world coordinate system defined and does that affect generalization? This is an important detail that's not discussed in the main text or supplement. Given this omission, it should also be clarified how the origin points of the arrows indicating the various axes are determined for in the wild data visualizations and how these arrows are projected from 2D to 3D.

Minor
L287 - Fine grain -> Fine grained

---

> ### Author Rebuttal · Authors · 2025-07-31
>
> Thank you for your supportive review and suggestions. Below we respond to the comments in **Weaknesses (W)** and **Questions (Q)**.
>
> ---
>
> ## W1 & Q1: Effect of Data Engine
>
> Below are comparison results using the same number of object assets from different sources (our data engine vs. Objaverse):
>
> | Assets Type | Assets Number | Objectron | Objectron | Ori_COCO | LINEMOD   | LINEMOD  | YCB-Video | YCB-Video |
> | ----------- | ------------- | --------- | --------- | -------- | --------- | -------- | --------- | --------- |
> |             |               | Med↓      | Acc15↑    | Acc↑     | Med↓      | Acc15↑   | Med↓      | Acc15↑    |
> | Objaverse   | 40K           | 25.05     | 54.3      | **74.8** | 10.70     | 69.8     | 15.49     | 72.5      |
> | Ours        | 40K           | **24.44** | **54.6**  | 74.6     | **10.16** | **74.1** | **7.28**  | **76.2**  |
>
> The comparison shows that even without considering scalability, our data engine has significant advantages in asset quality and annotation accuracy.
>
> ---
>
> ## W2 & Q2: Domain gap between synthetic and real data
>
> We have adopted two strategies to improve domain transfer:
>
> 1. We mixed annotated real images (i.e., ImageNet3D) into our training data.
> 2. We initialized our models with strong base models pre-trained on large-scale real data, which has been shown to significantly improve synthetic-to-real transfer in works such as Orient Anything v1, Depth Anything v2[1], and Marigold[2].
>
> Additionally, more domain gap mitigation methods can be explored. For example, we found that using advanced depth-conditioned image generation models can redraw rendered images into photorealistic ones while preserving object structure, which could be a promising direction for addressing domain gaps in the future.
>
> [1] Yang L, Kang B, Huang Z, et al. Depth anything: Unleashing the power of large-scale unlabeled data. (CVPR 2024)
>
> [2] Ke B, Obukhov A, Huang S, et al. Repurposing diffusion-based image generators for monocular depth estimation. (CVPR 2024)
>
> ---
>
> ## W3 & Q3: World coordinate system
>
> During rendering, we align the center of each 3D object's bounding box to the origin, so by definition, our orientation is aligned with the object's 3D box center.
>
> However, our model only outputs three Euler angles and does not predict the 3D box center. For visualization, we currently overlay the orientation arrow at the center of the object's 2D bounding box for quick illustration. More precise alignment with the origin would require combining advanced depth prediction or monocular 3D box detection models to estimate the object's 3D box center.
>
> ---

---

> > ### Comment · Reviewer_krXG · 2025-08-05
> >
> > Thank you for answering my rebuttal questions. It is good to directly see that the data engine has a direct positive effect on certain datasets, as well as understand the clarifications about how the visualizations are generated.
> >
> > Please include both the answers to Q1 and Q3 in the updated version of this paper. Explaining how the orientation arrows are visualized is especially important to get added as early as possible in the draft (this will greatly help the reader understand what's going on). The paper is good quality, the rebuttal response clarifies my questions, so I maintain my positive rating.

---

> > > ### Author Response · Authors · 2025-08-06
> > > **Thank you for your support!**
> > >
> > > We sincerely appreciate your kind support and valuable comments. In our final revision, we will further improve the paper by incorporating the valuable insights gained from the rebuttal discussions (e.g., our responses to Q1 and Q3). Please let us know if you have any further questions or suggestions; we would be pleased to address them.

---

### Official Review · Reviewer_k3h2 · 2025-07-03

**Clarity:** 4
**Significance:** 4
**Originality:** 4
**Rating:** 6
**Confidence:** 4

**Summary:**

This paper focuses on the problem of estimating object orientation from real world images. It builds on Orient Anything V1 and makes multiple improvements to the data collection and training pipeline to enhance its capabilities.

First, they use recent 3D generative models to collect a diverse dataset of objects much beyond the previous work on Orient Anything V1. Additionally, they also use some heuristics to annotate multiple front faces on the horizontal plane for objects which are symmetric.

They then train a model to predict multiple plausible orientations of the object using this dataset, which is a significant advance compared to Orient Anything V1 which predicted only one possible orientation. Further, unlike Orient Anything V1 the same model is also capable of estimating the relative rotation between a pair of frames.

**Questions:**

1. Can the method be extended to work on scenes with multiple objects? It would be interesting to see some results on this and whether the model would generalize. The test datasets contain only one object per scene.

2. I’m also curious if the method can work on arbitrary object classes which don’t exist in the training set. Is there a way to evaluate this?

**Ethical Concerns:**

["NO or VERY MINOR ethics concerns only"]

**Final Justification:**

My questions were sufficiently addressed during the discussion, so I will keep my original rating.

**Limitations:**

yes

**Quality:**

4

**Strengths And Weaknesses:**

**Strengths:**

1. The paper demonstrates impressive results across multiple tasks. For single-image absolute orientation estimation, the model outperforms Orient Anything V1 by a large margin. It also introduces two additional tasks—relative rotation estimation and horizontal-plane symmetry detection—and shows superior performance compared to strong baselines.

2. I found Orient Anything V1 to be quite limited in applicability due to its inability to handle symmetric objects or those with no clear front face (as illustrated by the skateboard example in Figure 1). This work addresses that limitation effectively by modeling all plausible front faces for such objects.

3. A particularly interesting contribution is the emergence of multiple front-face detection via simple statistical analysis of the predicted orientation distribution from a pretrained VLM trained on absolute orientation tasks. Despite the ill-posed nature of the problem, analyzing predicted pose distributions for each class over the dataset allows the model to detect multiple plausible front faces. This is a significant and impactful result, as there has been no clear method in the field for automatically identifying all valid orientations of arbitrary objects.

4. The paper also employs some clever design choices, such as pretraining VGGT and leveraging the camera pose token—which contains related information—for orientation prediction. This results in a highly capable and versatile model.

**Weaknesses:**

1. In Figure 1, the tasks depicted in the third column are difficult to follow. Specifically, the overlay of the two coordinate systems in the third row is hard to interpret, and it is unclear that different tasks are being illustrated.

2. Lines 23–24 are confusing; it’s not clear what it means for a model to have an intrinsic link to orientation yet not understand orientation. This could be clarified.

3. Line 46—“and sets new records”—sounds overly celebratory. A more neutral phrasing would be more appropriate.

---

> ### Author Rebuttal · Authors · 2025-07-31
>
> Thank you for your supportive review and suggestions. Below we respond to the comments in **Weaknesses (W)** and **Questions (Q)**.
>
> ---
>
> ## W1: Tasks depicted in the third column of Figure 1 are difficult to follow
>
> Our intention was to use semi-transparent arrows to indicate the orientation of the previous frame and annotate the predicted relative rotation. However, due to lack of explanation and insufficient distinction between arrows, this may have caused confusion. In future versions, we will add clarifications and use more distinguishable arrow styles (e.g., lighter colors) to differentiate them.
>
> ---
>
> ## W2: Confusing Lines 23-24
>
> We intended to convey that while Orient Anything can predict absolute orientation, it does not fully capture relative rotation, despite the intrinsic link between object orientation and rotation. We will modify and polish these statements to eliminate confusion.
>
> ---
>
> ## W3: Overly celebratory Line 46
>
> Thank you for your suggestion. In future versions, we will use more neutral phrasing, such as "robust performance" and "precise results."
>
> ---
>
> ## Q1: Extend to multiple objects
>
> As discussed in Orient Anything V1, by equipping with object detection and segmentation models, we can isolate each object in multi-object scenes and perform orientation prediction separately. This simple pipeline yields strong results.
>
> ---
>
> ## Q2: Application to arbitrary object classes not present in the training set
>
> We tested on some imaginary objects (e.g., fantasy animals or vehicles) that do not exist in our training set. Our model is able to make reasonable predictions based on common sense, such as object parts or proportions.
>
> Additionally, SynCLR [1], which also synthesizes text-image paired data from ImageNet21K class tags using LLMs and image generation models, demonstrates strong generalization to arbitrary categories. This may be due to the inherent diversity of image generation models. Our method contains similar category coverage as SynCLR, and thus can be expected to generalize similarly to arbitrary categories.
>
> [1] Tian Y, Fan L, Chen K, et al. Learning vision from models rivals learning vision from data. (CVPR 2024)
>
> ---

---

> > ### Comment · Reviewer_k3h2 · 2025-08-05
> > **thanks for the clarifications**
> >
> > Thanks for the clarifications, especially regarding the presentation in Figure 1 and the extension to multi-object scenes. That answers my question.
> >
> > I was also curious: do you plan to include the results on arbitrary categories in the paper? It would be great to see a few examples or a brief analysis of how the model handles such cases.

---

> > > ### Author Response · Authors · 2025-08-06
> > > **Thank you for your support!**
> > >
> > > We sincerely appreciate your kind support and valuable comments. In our final revision, we will further improve the paper by incorporating the valuable insights gained from the rebuttal discussions (e.g., the results and analysis on arbitrary categories). Please let us know if you have any further questions or suggestions; we would be pleased to address them.

---

### Official Review · Reviewer_13NV · 2025-07-07

**Clarity:** 4
**Significance:** 3
**Originality:** 3
**Rating:** 5
**Confidence:** 4

**Summary:**

This paper proposes a new model for single-view and multi-view viewpoint estimation and symmetry detection. The model builds on the existing Orient Anything model and extends that approach through proposing a novel data engine that leverage generative and VLM models. The data engine uses LLMs to generate detailed object descriptions for a large number of object types, text2mesh models to generate the 3D meshes, and VLMs to identify the front view (through a process of multiview generation, voting, and human calibration at the end). The proposed model starts with the VGGT model and trains it with the generated assets and annotation to estimate the object orientation and possible symmetries. The proposed approach achieves impressive performance on several datasets. The paper also details several ablations to evaluate impact of scale, weight initialization, and asset type.

**Questions:**

- How well does Orient Anything V1, VGGT, and RelPose perform on relative rotation estimation? While I think all 3 comparisons are important, VGGT seems to be the crucial one give the results shown in Table 4.
- The paper proposes a novel data engine and the resulted generated dataset. However, it's unclear from the text of the paper whether there are plans to release the data engine (code to generate this data) or the generated artifacts. The checklist asks the reader to refer to Sec 6, but it was unclear from Sec 6 what the plans are. Could you please clarify this? As the guidelines indicate, this is not a requirement but it does impact the contribution of the work to the community.


The following questions are less important and more of curiosities about how the model would perform under different edge cases. I think discussing them would improve the paper.
- The paper appears to make an assumption regarding rotation symmetries only existing around the vertical (or up) vector (L66). This assumption doesn't seem to always hold; eg, a strafish's front view is likely facing the star shape, resulting in the rotation being relative to the front vector. I am curious how you think the model would react to something like this? Similarly for a wheel or a log, both of which will have rotation symmetries around right/left or front/back vectors.
- Another example I am curious about is a sphere which would have 0 front faces (as well as no well defined up vector). This is discussed briefly in L70 as having no meaningful direction, but I am curious how the model would perform on such a case.

**Ethical Concerns:**

["NO or VERY MINOR ethics concerns only"]

**Final Justification:**

After looking at the reviews and the authors' responses, I would recommend the acceptance of this work. This paper addresses an interesting problem, proposes a holistic solution that combines improvements in representing the problem as well as a scalable data engine. The paper also analyzes the various components of the pipeline to understand the contribution of different model/data-engine components and design choices. As a result, I think this work would be valuable.

**Limitations:**

Yes.

**Quality:**

3

**Strengths And Weaknesses:**

**Strengths**
- The paper is very well written. I especially appreciated the nuance and care taken in explaining the challenges with rotation symmetry and what examples don't make sense, as well as the limitations of prior work. I also appreciated the discussion of the limitations of real data and what generative models can provide.
- The data engine is very well motivated and explained. The figures were especially helpful to understand what's going on. The manual verification and calibration also seems like a lot of work (L162-166).
- The experimental section is nicely presented and the ablations done make a lot of sense.

**Weaknesses**
- The paper mentioned that Orient Anything V1 struggled in the relative rotation estimation task (L108, L114), yet, it's not evaluated on the relative rotation estimation task.
- Similar to the previous point, VGGT (which is the basis for the proposed model) and RelPose [ref1] both seem like relevant comparisons for the relative pose estimation task. VGGT is incredibly relevant given that the results in Table 4 frame as a major contributor to the model's performance.
- The paper makes several claims about limitations in the literature that ignore existing work on rotation symmetry and relative pose estimation especially regarding lack of open-world scenarios and limitation to category-specific methods. There exists several papers that tackled those problems in the past. For example, Ref2 predicted a probability distribution for single image rotation that capture rotation symmetries. Similarly, Ref1 predicted a distribution for pairwise views that also accounted for rotation symmetries. In terms of relative rotation estimation, Ref1 has been applied in more open-world settings. Furthermore, Ref3-4 were earlier works for relative pose estimation that did not make category/object specific assumptions.


References:
- [Ref1] Zhang et al. "Relpose: Predicting probabilistic relative rotation for single objects in the wild." European Conference on Computer Vision. Cham: Springer Nature Switzerland, 2022.
- [Ref2] Murphy et al. "Implicit-PDF: Non-Parametric Representation of Probability Distributions on the Rotation Manifold." International Conference on Machine Learning. PMLR, 2021.
- [Ref3] El Banani et al. "Novel Object Viewpoint Estimation Through Reconstruction Alignment." 2020 IEEE/CVF Conference on Computer Vision and Pattern Recognition (CVPR). IEEE, 2020.
- [Ref4] Park et al. "Latentfusion: End-to-end differentiable reconstruction and rendering for unseen object pose estimation." Proceedings of the IEEE/CVF conference on computer vision and pattern recognition. 2020.

---

> ### Author Rebuttal · Authors · 2025-07-31
>
> Thank you for your supportive review and suggestions. Below we respond to the comments in **Weaknesses (W)** and **Questions (Q)**.
>
> ---
>
> ## W1 & W2 & Q1: More baselines in relative rotation estimation tasks
>
> We report the median error (Med↓) of more baselines, Orient Anything v1, VGGT, and two versions of Relpose (co3dv1 and co3dv2), on the relative rotation prediction benchmark as follows:
>
> | Model              | LINEMOD | YCB-Video | OnePose++ | OnePose |
> |--------------------|---------|-----------|-----------|---------|
> | **POPE's Sampling**|         |           |           |         |
> | Gen6D              | 44.86   | 54.48     | 35.43     | 17.78   |
> | LoFTR              | 33.04   | 19.54     | 9.01      | 4.35    |
> | POPE               | 15.73   | 13.94     | **6.27**  | **2.16**|
> | VGGT               | 16.64   | 10.97     | 17.76     | 14.08   |
> | Relpose-co3dv1     | 16.74   | 13.53     | 14.54     | 11.37   |
> | Relpose-co3dv2     | 14.27   | 9.76      | 12.99     | 9.52    |
> | OriAny. V1         | 22.67   | 16.27     | 13.32     | 22.30 |
> | OriAny. V2         | **9.45**| **6.53**  | 7.62      | 8.53    |
> | **Random Sampling**|         |           |           |         |
> | POPE               | 98.03   | 41.88     | 88.21     | 45.73   |
> | VGGT               | 83.96   | 20.75     | 52.18     | 40.30   |
> | Relpose-co3dv1     | 100.05  | 43.17     | 81.35     | 76.67   |
> | Relpose-co3dv2     | 95.49   | 31.21     | 83.21     | 47.48   |
> | OriAny. V1         | 86.63   | 43.30     | 26.88     | 75.19   |
> | OriAny. V2         | **32.08**| **13.78**| **12.94** | **16.29**|
>
> ---
>
> ## W3: Lack of discussion on existing work on rotation symmetry and relative pose estimation
>
> Thank you for your suggestion. We will revise our wording and add discussions of these papers in the updated version. In particular, Relpose [Ref1] demonstrates better generalization to open-world scenarios, and we further include it as one of the baselines for the relative pose estimation task. The comparison results are provided in **our response to your W1 & W2 & Q1 above**.
>
> ---
>
> ## Q2: Open-sourcing the data engine and resulted dataset
>
> We will do our best to open-source our work. Each part of our data engine is based on open-source models or methods, and we will organize and release our data engine pipeline. However, for the resulted datasets, the situation is more complex: our 3D object assets were generated using Hunyuan-3d-2.0, and we need to further confirm the license with the Hunyuan team before deciding whether to release large-scale generated 3D assets.
>
> ---
>
> ## Q3: Only considering vertical rotation symmetries
>
> To simplify the problem, we only considered vertical rotation symmetry, which covers most cases. However, as you pointed out, some objects may exhibit rotational symmetry along the left-right or front-back axes. We believe that multi-view rendering and prediction voting along these axes may help capture such symmetries.
>
> ---
>
> ## Q4: How does the model perform for spheres
>
> Thank you for raising this point. We tested several sphere cases and observed some interesting phenomena. For spheres placed on a plane (with background), the model predicts a pitch angle relative to the supporting plane (i.e., the object's up vector is perpendicular to the plane). For spheres without a background, the model does indeed predict a random 'up' direction, which makes sense because any angle is valid. This aligns with human perception and observation.
>
> ---

---

> > ### Comment · Reviewer_13NV · 2025-08-04
> >
> > Thank you for responding to my the points I raised and even addressing some of the edge cases I was curious about.
> >
> > I have also read the other reviews and the author's responses. Overall, I find the author's responses to the raise concerns acceptable. Since reviewer  tGZ7 is the only one who was leaning for rejection, I am curious if they feel that their concerns were addressed.

---

> > > ### Author Response · Authors · 2025-08-05
> > > **Thank you for your support!**
> > >
> > > We sincerely appreciate your kind support. In our final revision, we will further enhance the paper by incorporating the valuable insights gained from the rebuttal discussions. Thank you once again for your guidance. Please let us know if you have any further questions or suggestions; we would be pleased to address them.

---

### Decision · Program_Chairs · 2025-09-17

**Decision:**

Accept (spotlight)

**Comment:**

Following the discussion phase, all reviewers recommended acceptance (1 Strong Accept, 3 Accepts), noting that the paper is interesting, well-written, novel, thoroughly evaluated, and presents impressive results. The rebuttal addressed many of the reviewers’ concerns—for example, by clarifying technical details, committing to expand the related work discussion, and adding new experiments on initialization. As a result, the ACs have decided to accept the paper. Please take the reviewers’ feedback into account when preparing the camera-ready version.